# LEARNING TO DESCRIBE SCENES WITH PROGRAMS

**Yunchao Liu**[*]
IIIS, Tsinghua University

**Zheng Wu**
MIT CSAIL, Shanghai Jiao Tong University

**Daniel Ritchie**
Brown University

**William T. Freeman**
MIT CSAIL, Google Research

**Joshua B. Tenenbaum**
MIT CSAIL

**Jiajun Wu**
MIT CSAIL

## ABSTRACT

Human scene perception goes beyond recognizing a collection of objects and their pairwise relations. We understand higher-level, abstract regularities within the scene such as symmetry and repetition. Current vision recognition modules and scene representations fall short in this dimension. In this paper, we present *scene programs*, representing a scene via a symbolic program for its objects, attributes, and their relations. We also propose a model that infers such scene programs by exploiting a hierarchical, object-based scene representation. Experiments demonstrate that our model works well on synthetic data and transfers to real images with such compositional structure. The use of scene programs has enabled a number of applications, such as complex visual analogy-making and scene extrapolation.

## 1 INTRODUCTION

When examining the image in Figure 1a, we instantly recognize the shape, color, and material of the objects it depicts. We can also effortlessly imagine how we may extrapolate the set of objects in the scene while preserving object patterns (Figure 1b). Our ability to imagine unseen objects arises from holistic scene perception: we not only recognize individual objects from an image, but naturally perceive how they should be organized into higher-level structure (Rock & Palmer, 1990).

Recent AI systems for scene understanding have made impressive progress on detecting, segmenting, and recognizing individual objects (He et al., 2017). In contrast, the problem of understanding high-level, abstract relations among objects is less studied. While a few recent papers have attempted to produce a holistic scene representation for scenes with a variable number of objects (Ba et al., 2015; Huang & Murphy, 2015; Eslami et al., 2016; Wu et al., 2017), the relationships among these objects are not captured in these models.

The idea of jointly discovering objects and their relations has been explored only very recently, where the learned relations are often in the form of interaction graphs (van Steenkiste et al., 2018; Kipf et al., 2018) or semantic scene graphs (Johnson et al., 2015), both restricted to pairwise, local relations. However, our ability to imagine extrapolated images as in Figure 1 relies on our knowledge of long-range, hierarchical relationships among objects, such as how objects are grouped and what patterns characterize those groups.

In this paper, we aim to tackle the problem of understanding higher-level, abstract regularities such as repetition and symmetry. We propose to represent scenes as *scene programs*. We define a domain-specific language for scenes, capturing both objects with their geometric and semantic attributes, as well as program commands such as loops to enforce higher-level structural relationships. Given an image of a complex scene, we propose to infer its scene program via a hierarchical bottom-up approach. First, we parse the image into individual objects and infer their attributes, resulting in the *object representation*. Then, we organize these objects into different groups, i.e. the *group representation*, where objects in each group fall into the same program block. Finally, we describe each group with a program, and combine these programs to get the *program representation* for the entire scene.

---

[*]This work was done when Yunchao Liu was a visiting student at MIT CSAIL.

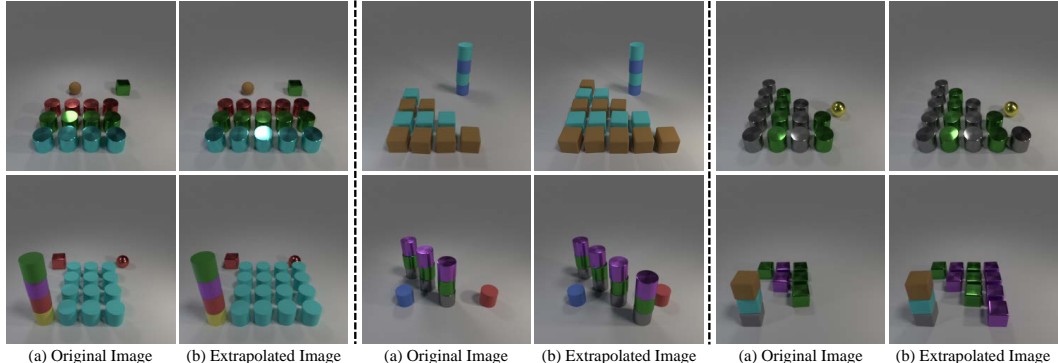

| (a) Original Image | (b) Extrapolated Image | (a) Original Image | (b) Extrapolated Image | (a) Original Image | (b) Extrapolated Image |

Figure 1: High-level scene understanding. Given original image (a), we are able to imagine unseen objects based on the structural relations among existing objects, resulting in extrapolated image (b).

Our model applies deep neural networks for each stage of this process and is able to generate programs describing the input image with high accuracy. When testing on scenes that are more complex than those used for training, our hierarchical inference process achieves better generalization performance than baseline methods that attempt to infer a program directly from the image. Our model is also able to handle ambiguity, generating multiple possible programs when there is more than one way to describe the scene. Furthermore, our method generalizes to real-world images without any additional supervised training programs; only the low-level object detection module must be re-trained. Finally, we demonstrate how our model facilitates high-level image editing, as users can change parameters in the inferred program to achieve the editing effects they want more efficiently. We show examples of such image edits, including extrapolations such as the one in (Figure 1b), on both synthetic and photographic images.

Our contributions are therefore three-fold:

1. We propose scene programs: a new representation for scenes, drawing insights from classic findings in cognitive science and computer graphics.
2. We present a method for inferring scene programs from images using a hierarchical approach (from objects to groups to programs).
3. We demonstrate that our model can achieve high accuracy on describing both synthetic and constrained real scenes with programs. Combined with modern image-to-image translation methods, our model generates realistic images of extrapolated scenes, capturing both high-level scene structure and low-level object appearance.

## 2 RELATED WORK

**Describing Images with Programs** Ellis et al. (2018) performs a similar task as ours where hand-drawn images of 2D geometry primitives are converted to high-level programs. This work uses a constraint-based SAT solver to perform program search and is much slower than neural network models. IM2LATEX (Deng et al., 2017) de-renders images into low-level LaTeX markup using a neural network, while our work discovers high-level programs from an image of objects. SPIRAL (Ganin et al., 2018) uses reinforcement learning to infer a sequence of low-level drawing commands that can reproduce an image, meanwhile learning a distribution from which images can be sampled. Beltramelli (2018) learns to convert GUI images to markup-like code. Unlike these papers, our model performs program induction in 3D and infers high-level structural patterns both in object layout and color.

**Describing the Structure of 3D Shapes and Scenes** Beyond 2D images, prior work in vision and graphics has attempted to infer high-level structure from 3D objects and 3D scenes. The most relevant to our approach are those that extract a so-called *symmetry hierarchy*, in which 3D geometry is hierarchically grouped by either attachment or symmetric relationships (Wang et al., 2011). This

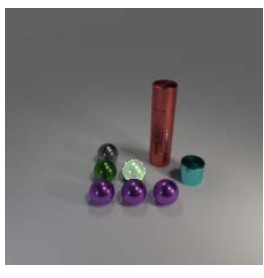 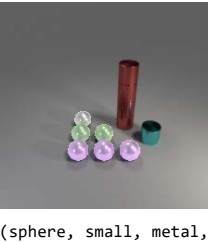 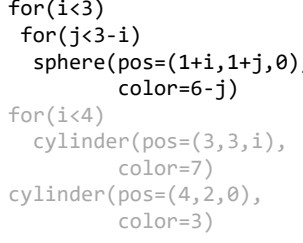

```
for(i<3)
 for(j<3-i)
  sphere(pos=(1+i,1+j,0),
         color=6-j)
for(i<4)
  cylinder(pos=(3,3,i),
          color=7)
cylinder(pos=(4,2,0),
         color=3)
```

(sphere, small, metal, green, x=2, y=2, z=0) …

(a) Input image
(b) Object parsing & group detection
(c) Program

Figure 2: Our model for visual program synthesis. (a) The input is an image consisting of multiple objects with ordered arrangements. We also perform instance segmentation to get object masks. (b) We use two vision models to extract object attributes and predict object groups, respectively. (c) These representations are then sent to a sequence model to predict the program.

representation has been used to train generative models of 3D shapes (Li et al., 2017) and indoor 3D scenes (Li et al., 2019), as well as to infer a hierarchical bounding box structure from a single image of a 3D shape (Niu et al., 2018). Our program representation bears some resemblance to the symmetry hierarchy, but it generalizes to repetitive patterns beyond symmetries and also models patterns in object visual attributes (*e.g.*, color). CSGNet (Sharma et al., 2018) learns to parse shapes with a set of primitive and arithmetic commands; Tulsiani et al. (2017) parses shapes into an assembly of geometric primitives. In this paper, we focus on learning the high-level scene regularities described by loop structures.

**Neural Program Synthesis** In general, a program synthesis model outputs an explicit program by learning from examples. Recent works on neural program synthesis include R3NN (Parisotto et al., 2017) and RobustFill (Devlin et al., 2017), which perform end-to-end program synthesis from input/output examples. Bunel et al. (2018) goes beyond the pure supervised learning setting and improves performance on diversity and syntax by leveraging grammar and reinforcement learning. These models synthesize programs based on input/output pairs, which is different from our setting, where a program is generated to describe an input image. In the vision domain, Sun et al. (2018) learns decision strategies represented as programs from demonstration videos, while we focus on describing the complex correlations among objects in static scenes.

## 3 METHOD

Our model combines vision and sequence models via structured representations. An object parser predicts the segmentation mask and attributes for each object in the image. A group recognizer predicts the group that each object belongs to. Finally, a program synthesizer generates a program block for each object group. Figure 2 shows an example of synthesizing programs from an input image, where a sphere is selected at random (highlighted) and the group that this object belongs to is predicted, which consists of six spheres. Then the program for this group (highlighted) is synthesized.

### 3.1 A DOMAIN-SPECIFIC LANGUAGE (DSL) FOR SCENES

In order to constrain the program space to make it tractable for our models, we introduce human prior on scene regularities that can be described as programs. More specifically, we introduce a Domain Specific Language (DSL) which explicitly defines the space of our scene programs. We present the grammar of our DSL in Table 1, which contains 3 primitive commands (`cube`, `sphere`, `cylinder`) and 2 loop structures (`for`, `rotate`). The positions for each object are defined as affine transformations of loop indices, while the colors are more complicated functions of the loop indices, displaying alternating (modular) and repeating (division) patterns.

Furthermore, since the DSL allows unbounded program depth, we define *program blocks* to further reduce complexity. Each type of program block is an production instance of the Statement token, and

| | | |
|---:|:---:|:---|
| Program | $\rightarrow$ | Statement; $\cdots$; Statement |
| Statement | $\rightarrow$ | `cube`(pos=Expression1, color=Expression2) |
| Statement | $\rightarrow$ | `sphere`(pos=Expression1, color=Expression2) |
| Statement | $\rightarrow$ | `cylinder`(pos=Expression1, color=Expression2) |
| Statement | $\rightarrow$ | `for`$(0 \leq$ Var1 $<$ Expression1$)\{$Program$\}$ |
| Statement | $\rightarrow$ | `rotate`$(0 \leq$ Var1 $<$ Expression1, start=$Z$, center=$(Z, Z, Z))\{$Program$\}$ |
| Expression1 | $\rightarrow$ | $Z \times$ Var1 $+ \cdots + Z \times$ Var1 $+ Z$ |
| Expression2 | $\rightarrow$ | $Z \times$ Var2 $+ \cdots + Z \times$ Var2 $+ Z$ |
| Var1 | $\rightarrow$ | a free variable |
| Var2 | $\rightarrow$ | Var1 $\mid$ Var1 $\% Z \mid$ Var1 $/ Z$ |
| $Z$ | $\rightarrow$ | integer |

Table 1: Grammar of the scene program. Primitive commands (`cube`, `sphere`, `cylinder`) can be placed inside loop structures, where the position and color of each object are determined by the loop indices.

objects that belong to the same block form a *group*. For example, in this work the program blocks include single objects, layered for loops of depth $\leq 3$, and single-layer rotations of $\leq 4$ objects.

## 3.2 OBJECT PARSING

Following the spirit of *The Trace Hypothesis* (Ellis et al., 2018), we use object attributes as an intermediate representation between image space and structured program space. Parsing individual objects from the input image consists of two steps: mask prediction and attribute prediction. For each object, its instance segmentation mask is predicted by a Mask R-CNN (He et al., 2017). Next, the mask is concatenated with the original image, and sent to a ResNet-34 (He et al., 2015) to predict object attributes. In our work, object attributes include shape, size, material, color and 3D coordinates. Each attribute is encoded as a one-hot vector, except for coordinates. The overall representation of an object is a vector of length 18. The networks are trained with ground truth masks and attributes, respectively. For the attribute network, we minimize the mean-squared error between output and ground truth attributes.

## 3.3 GROUP DETECTION

When we identify a distinct visual pattern, we first know which objects in the image form the pattern before we can tell what the pattern is. Motivated by this idea, we develop a group recognizer that tells us which objects form a group that can be described by a single program block. The group recognizer works after mask prediction is performed, and answers the following specific question: given an input object, which objects are in the same group with this object?

The input to the model consists of three parts: the original image, the mask of the input object, and the mask of all objects. These three parts are concatenated and sent to a ResNet-152 followed by fully connected layers. The output contains two parts: a binary vector $g$ where $g[i] = 1$ denotes object $i$ in the same group with the input object, and the category $c$ of the group, representing the type of program block that this group belongs to. The network is trained to minimize the binary cross entropy loss for group recognition, and the cross entropy loss for category classification.

## 3.4 NEURAL PROGRAM SYNTHESIS

With the object attributes and groups obtained from the vision models, the final step in our model is to generate program sequences describing the input image. Since we have already detected object groups, what remains is to generate a program block for each group. For this goal we train a sequence to sequence (seq2seq) LSTM with an encoder-decoder structure and attention mechanism (Luong et al., 2015; Bahdanau et al., 2015). The input sequence is a set of object attributes that form a group, which are sorted by their 3D coordinates. The output program consists of two parts: program tokens are predicted as a sequence as in neural machine translation, and program parameters are predicted by a MLP from the hidden state at each time step. At each step, we predict a token $t$ as well as a

---

**Algorithm 1:** Combining group prediction with program synthesis

---

**Result:** a program sequence $P$
**Input:** a set of object attributes $O$;
**while** *O is not empty* **do**
  randomly choose $o_i \in O$;
  predict the group that contains $o_i$, indexed by $G$;
  also predict the group category $c$;
  get attributes of objects that belong to the group, $A = \{o_j | j \in G\}$;
  remove $A$ from $O$;
  send $A, c$ to program synthesizer, get program $p$;
  add $p$ to $P$;
**end**

---

parameter matrix $P$, which contains predicted parameters for all possible tokens. Then we use $P[t]$ as the output parameter for this step.

Since the program synthesizer only works for a single group, a method for combining the group prediction with program synthesis is needed. Consider the simplest case where we randomly choose an object and describe the group it belongs to. This procedure is described in Algorithm 1. In practice, by default we sample 10 times and stop when a correct program is generated. Here correct means that we can recover the scene attributes successfully by executing the program.

## 4 EXPERIMENTS

We perform several experiments on synthetic scene images, including quantitative comparison with baseline methods and further extensions and applications. We further demonstrate our model's ability to generalize to real images with a small amount of hand-labeled supervision which is only at the object level. We also apply our method to other tasks, specifically image extrapolation and visual analogy-making, on both synthetic and real images.

### 4.1 DATASET

We create a synthetic dataset of images rendered from complex scenes with rich program structures. Figure 3 displays some examples drawn from the dataset. These images are generated by first sampling scenes and then rendering using the same renderer as in CLEVR (Johnson et al., 2017). Each scene consists of a few groups, where objects in the same group can be described by a program block. The groups are sampled from predefined program primitives with multi-layered translational and rotational symmetries. Further, we also incorporate rich color patterns into the primitives.

Our synthetic dataset includes annotations on object attributes and programs. We train and test the models on two synthetic datasets, REGULAR and RANDOM, each containing 20,000 training and 500 test images, where each image has at most 2 groups of multiple objects, in addition to many groups of a single object. In the REGULAR dataset (Figure 3a), the objects are placed on a grid and have discrete coordinates. We increase the scene complexity by adding randomness in the RANDOM dataset (Figure 3c), where objects are placed uniformly at random with continuous coordinates and different sizes.

### 4.2 VISUAL PROGRAM SYNTHESIS

**Setup.** We present evaluation results on our synthetic dataset introduced above. We compare with an ablated version of our full model, where we use a simple search-based heuristic grouping method (HG) which does not require any training or any knowledge of the program patterns. The details of this method are presented in Appendix A. We also compare with two baselines. One removes group recognition and instead synthesizes programs from all object attributes (derender-LSTM). Another directly synthesizes programs from the input image in an end-to-end manner (CNN-LSTM). The model uses a CNN as encoder and a LSTM with attention as decoder. We use the same network

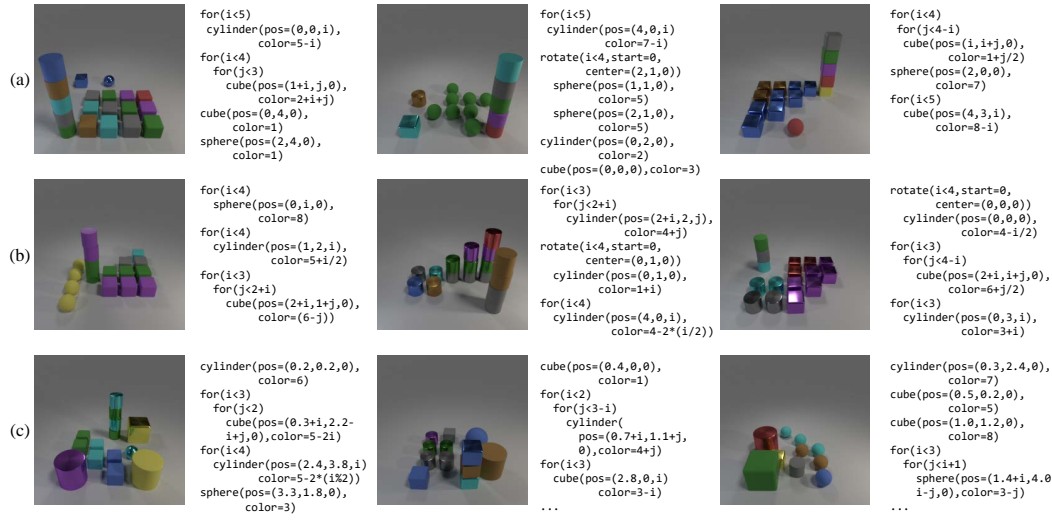

Figure 3: Qualitative results for visual program synthesis. (a) Results on the REGULAR test set, where all objects are placed on a grid. (b) Results on the generalization test set which contains more complex scenes. (c) Results on the RANDOM test set, where objects have different sizes and the groups are placed with random continuous coordinates.

| Model | Token (%) | Param. (MSE) | Test (%) | Generalization (%) | Random (%) |
|---|---|---|---|---|---|
| ours (full) | **99.5** | **0.014** | **96.6** | **70.0** | **97.6** |
| ours (HG) | **99.5** | **0.014** | 92.4 | 63.0 | 94.8 |
| derender-LSTM | 97.5 | 0.080 | 87.6 | 14.0 | 64.4 |
| CNN-LSTM | 98.9 | 0.043 | 92.3 | 48.0 | 70.7 |

Table 2: Comparing program synthesis performance with baseline methods. Evaluation metrics include program token accuracy, parameter MSE loss, and scene reconstruction accuracy on the test sets.

architecture as in attention-based neural image captioning (Xu et al., 2015), except that the decoder predicts a token as well as a parameter matrix at each time step.

We evaluate the models on both the REGULAR and the RANDOM test sets. For evaluation on generalization, we also create an additional test set of 100 images, where each image contains three groups of multiple objects. These images are more complex and harder to describe than those in training.

**Results.** Figure 3 includes qualitative results generated by our model on all three test sets. Our model generates accurate results in the REGULAR setting and is able to recognize the two groups from neighbouring objects (Figure 3a). Although trained on images with two groups, our model can perform well when tested on images with three groups (Figure 3b). When objects are placed at random, our model can accurately recognize which objects form a regular pattern and describe them with programs (Figure 3c). For quantitative evaluation, we compute program token accuracy and parameter loss, defined as the percentage of correctly predicted tokens and the mean-squared error of parameter prediction, respectively. To evaluate the global performance of the generated program, we also compute reconstruction accuracy of the programs, defined as the percentage of programs that correctly reconstruct the original image. The reconstruction accuracy is evaluated on all three test sets.

We present the test results in Table 2, where our model outperforms baseline methods in each of the metrics, and achieves good performance on generalization. Note that the deep grouping model outperforms the simple heuristic grouping method, as it learns from the data distribution specified by

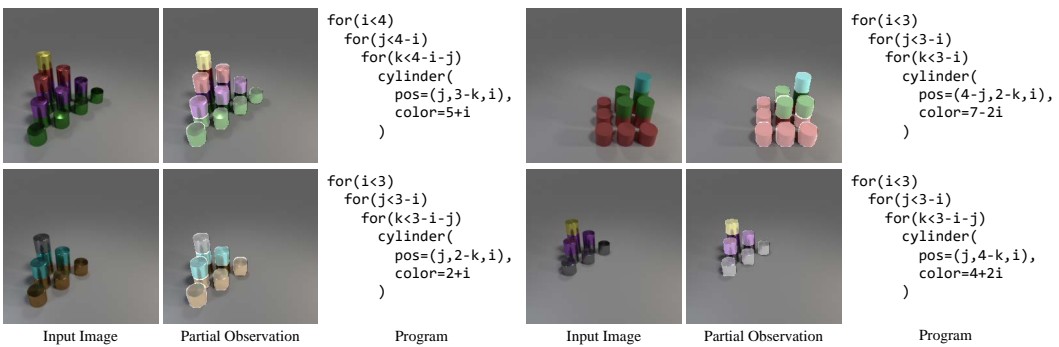

Figure 4: Generating multiple possible programs.

```
                                    for(i<4)                                for(i<3)
                                      for(j<4-i)                              for(j<3-i)
                                        for(k<4-i-j)                            for(k<3-i)
                                          cylinder(                              cylinder(
                                            pos=(j,3-k,i),                          pos=(4-j,2-k,i),
                                            color=5+i                               color=7-2i
                                          )                                       )

                                    for(i<3)                                for(i<3)
                                      for(j<3-i)                              for(j<3-i)
                                        for(k<3-i-j)                            for(k<3-i-j)
                                          cylinder(                              cylinder(
                                            pos=(j,2-k,i),                          pos=(j,4-k,i),
                                            color=2+i                               color=4+2i
                                          )                                       )
```

Input Image   Partial Observation   Program   Input Image   Partial Observation   Program

Figure 5: Inferring programs from partial observations. *(Input Image)* The input image contains objects that are fully or mostly occluded. *(Partial Observation)* Output of Mask R-CNN where we discard mask proposals that are too small. The highlighted objects form the observation of our model. *(Program)* Despite the noisy and incomplete input, our model can accurately predict programs that describe the image.

our program space. Also note that our model performs better on RANDOM, while the baselines do not perform as well. This is because our group detection model discovers groups among randomly placed objects better than among regularly placed objects, as it is easier to rule out outsiders when they look random.

**Tackling ambiguous input.** While our model can generate program representations for images with high accuracy, it can also generate multiple possible programs when the input is ambiguous. Figure 4 shows an example where the red group can be described by either a two-layer for loop or a rotation of 4 objects. Our hierarchical method allows explicit specification of group category. When executing Algorithm 1, instead of selecting the most confident group category, we search top 3 proposals, and execute the synthesized program block to decide if each proposal corresponds to a possible correct program. Figure 4 demonstrates programs generated by our model, while the baseline methods tend to collapse to one possible answer and is unable to generate others.

**Program synthesis from partial observations.** Our model can also handle scenes where there are invisible (or hardly visible) objects. Figure 5 demonstrates how our model operates on these scenes. Given an input image, we generate object instance masks and remove those with area below a certain threshold, so that the remaining objects can be correctly recognized. These objects form the partial observation of our model, from which the program synthesizer generates a program block which correctly describes the scene, including (partially) occluded objects. The flexibility of the neural program synthesizer allows us to recognize the same program pattern given different partial observations. Consider the two examples at the bottom of Figure 5. They have different set of observations (8 and 6 objects on bottom left and right, respectively) due to the different distances, and our model is able to correctly recognize both of them.

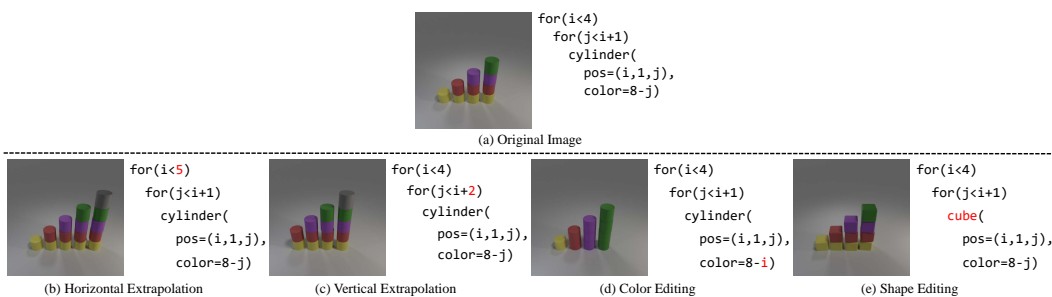

Figure 6: Image Editing. Our model can be applied to edit images by inferring programs (a) and then operate on program space. Examples include image extrapolation (b, c) and attribute editing (d, e).

### 4.3 Image Editing

**Image editing via program representation.** With the expressive power of the program representation, our model can be applied to tasks that require a high-level structural knowledge of the scene. For example, when an image lies within the space defined by our DSL, it can be efficiently edited using the program representation generated by our model. Figure 6 shows some examples of image editing, where the input image (Figure 6a) is represented by a program. Users can then edit the program to achieve the preferred editing effects. The edited program is sent to a graphics engine to render the new image. The structural form of our program representation allows various types of high-level editing, including spacial extrapolation (Figure 6b, c), changing color patterns (Figure 6d) and shapes (Figure 6e). Each of the four examples requires only one edit in the program, while using the traditional object representation, users have to change objects one at a time, averaging 6.25 edits per image.

**Real image extrapolation.** An advantage of our method which uses object attributes as a connection between vision and program synthesis is to generalize to real images. Since our neural program synthesizer is independent from visual recognition, only the vision systems need to be retrained for our entire model to work on real images.

Figure 7a shows images of LEGO blocks shot from a camera in real-world settings. We create a dataset of 120 real images, where we use 90 for training, 10 for validation, and 20 for testing. To adapt our model to generate programs for these images, we first pretrain on a synthetic dataset of 4,000 images rendered by a graphics engine. Then we fine-tune the model on 90 real images with labeled masks and attributes. The vision system is then linked with the pretrained program synthesizer which does not require any fine-tuning. Even with a small amount of real data for fine-tuning, our model generalizes well and correctly predicts the programs for each test image.

Furthermore, the image editing techniques introduced above can also be applied to such real images. Here we present an experiment on real image extrapolation. Given an input image, we generate the program describing the image and also extract object patches with Mask R-CNN. The program is extended by increasing the iteration number, which is a simple way of "imagining" what could be the next given a sequence of observations.

Our original method uses a graphics engine to render new images from edited programs (Figure 6), which is not applicable for real images. For this purpose, we use pix2pix (Isola et al., 2017) as an approximate neural renderer. After program inference, we execute the edited program and retrieve newly added object masks. These masks can be computed using camera parameters and 3D coordinates, while here we use retrieval for simplicity. All of the patches are pasted on a white background, and then sent to pix2pix to generate realistic background and lighting. Figure 7c displays the editing results. The edited images preserve object appearances in the original images, and also fix the errors made by mask prediction (small white gaps in Figure 7b) and contain realistic-looking shadows.

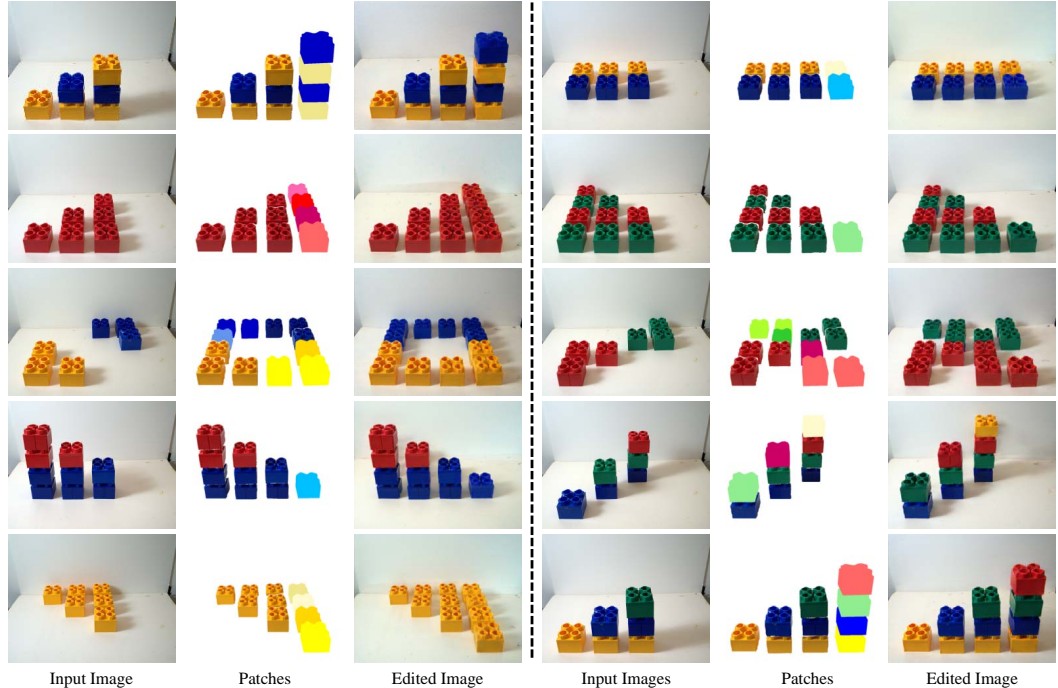

|         |         |              |              |         |             |
|---------|---------|--------------|--------------|---------|-------------|
| Input Image | Patches | Edited Image | Input Images | Patches | Edited Image |

Figure 7: Generalizing to real images. (a) Input image which is described by a program generated by our model. (b) The object patches in the original image are extracted using Mask R-CNN, while new objects are inferred by modifying the program iteration number and added as masks. (c) The edited image is rendered by pix2pix.

| Model | Synthetic | Real |
|-------|-----------|------|
| Autoencoder | 5.17 | 10.19 |
| Ours | **3.87** | **7.05** |

Table 3: Average L2 distance between ground truth images and model outputs for visual analogy making experiment.

## 4.4 VISUAL ANALOGY MAKING

Besides representing images for efficient editing, scene programs can also be used as encoded images. For example, the distance in program space can also be applied to model similarity between images, which is already introduced by (Ellis et al., 2018). Motivated by this idea, we consider visual analogy making (Reed et al., 2015), where an input image is converted to a new image given other reference images. We introduce a setting where the reference is an image pair and ask the intuitive question, *if B follows A, then what should follow C?*

Here we use a simple solution based on representation distance. More specifically, for an encoder $R$ and an input image $c$ with reference pair $(a, b)$, we set $R(d) = R(c) + R(b) - R(a)$ and decode $R(d)$ to get the output. In our case, the encoder is our program synthesis model, while we use pix2pix as a neural decoder. In order to perform arithmetic operations, the program is represented as a matrix, where each line starts with a token followed by parameters (see Appendix A.3 for details). We compare our model with an autoencoder (Hinton & Salakhutdinov, 2006). The autoencoder we adopt takes an input image of size $256 \times 256$, encodes the input into a 256-dimensional vector and then decodes the encoded vector back to original image size.

Figure 8 shows qualitative results of the visual analogy making task. Using our program representation, our model generates perceptually plausible results (Figure 8e). While the autoencoder

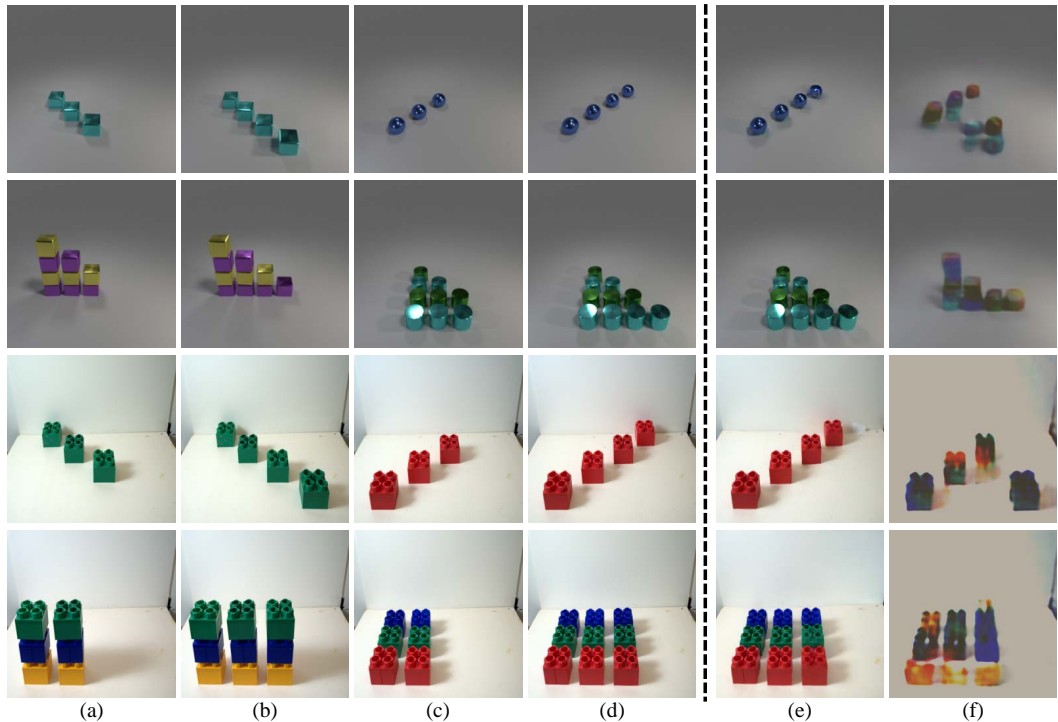

(a)      (b)      (c)      (d)      (e)      (f)

Figure 8: Visual Analogy Making. Given example image pairs (a), (b), the input image (c) is encoded with a representation, which is edited according to the example image pair. The edited representation is then decoded into a new image by our model (e) and an autoencoder (f), respectively. (d) shows analogy making result made by human.

can sometimes correctly change the number of objects, it fails to preserve the layout arrangements (Figure 8f). We also compute the average L2 distance between model output and ground truth images made by humans. Table 3 shows that our model generates images that are closer to the ground truth than the baseline.

## 5 CONCLUSION

We propose scene programs as a structured representation of complex scenes with high-level regularities. We also present a novel method that infers scene programs from 2D images in a hierarchical bottom-up manner. Our model achieves high accuracy on a synthetic dataset and also generalizes to real images. The representation power of programs allows our model to be applied to other tasks in computer vision, such as image editing and analogy making, on both synthetic and photographic images.

**Acknowledgements.** We thank Jiayuan Mao for insightful discussions and anonymous reviewers for their helpful feedback. This work was supported in part by NSF #1231216, NSF #1447476, NSF #1753684, ONR MURI N00014-16-1-2007, Facebook, and the Yao Class Exchange Program at IIIS, Tsinghua University.

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

# A    IMPLEMENTATION DETAILS

## A.1    SCENE CONFIGURATION

For synthetic data rendering, we use essentially the same settings as in CLEVR (Johnson et al., 2017). The objects are in two sizes (radius 0.4, 0.7), three shapes (`sphere`, `cube`, `cylinder`), two materials (`metal`, `rubber`), and eight colors (`blue`, `brown`, `cyan`, `gray`, `green`, `purple`, `red`, `yellow`). We represent the colors as numbers 1-8 in the scene programs as shown in Figure 3.

In the REGULAR setting, objects are placed on a $5 \times 5 \times 5$ grid, with integer coordinates in 0-4, which means that the spacial gap between objects is a constant 1. We only use objects with the smaller size. Then we jitter each object position by a random noise sampled uniformly from $[-0.03, 0.03]$. These random noises are used for visual diversity, and is ignored in the scene program.

In the RANDOM setting, objects have both large and small sizes, and we use continuous coordinates in $[0, 4]$. When sampling a program block, the spacial gap between neighboring objects in the same group is still the constant 1, while the entire group is shifted by a continuous random amount. Finally, each object is also independently jittered by a random noise sampled uniformly from $[-0.03, 0.03]$.

## A.2    HEURISTIC GROUPING

We present the detailed method for heuristic grouping (HG) in Algorithm 2. Here $d(o, G)$ denotes the minimum Euclidean distance from $o$ to any object in $G$. During testing we will sample the distance threshold $\varepsilon$ multiple times.

## A.3    DATA FORMAT FOR SCENE PROGRAMS

In this Section we introduce the data format for the proposed scene programs. In short, a program is represented as a matrix, where each row contains a program command, which is a program token followed by its parameters. In our work, a program block is represented as a matrix of size $N \times 14$ where $N$ is the number of program commands. In order to unify the data format of the programs specified by our DSL defined in Table 1, we divide the 14 numbers into four parts: program token (index 0), iteration arguments (index 1-3), position arguments (index 4-6) and color arguments (index

---

**Algorithm 2:** A simple heuristic grouping algorithm

---

**Result:** a group of objects $G$
**Input:** a set of object attributes $O$, an object $o \in O$;
$G = \{o\}$;
uniformly sample $\varepsilon$ from $[1, \sqrt{2}]$;
**while** *True* **do**
    **for** $o_i \in O$ **do**
        **if** $o_i \notin G$ *and* $d(o_i, G) < \varepsilon$ *and* $o_i$ *has the same shape as* $o$ **then**
             add $o_i$ to $G$
        **end**
    **end**
    **if** $G$ *did not change in this iteration* **then**
        **return** $G$
    **end**
**end**

---

7-13). We give an explicit example as shown in Figure 9. The matrix representation allows direct arithmetic operations in program space, which enables the application of scene programs in image analogy making (Figure 8).

The evaluation of output program is different under different scene configurations. In the REGULAR setting, every program argument is an integer, so we round the output program to the nearest integer and calculate its accuracy. In the RANDOM setting, since the position arguments are continuous, we allow a small error (0.2) for them and treat the other arguments as integers.

```
for(i<3)
  for(j<2)
    cube(pos=(0,1,0)          [[4, 0, 0, 3, 1, -1, 0, 0, 0, 0, 0, 0, 0],
              +i*(1,-1,0)       [4, 0, 0, 2, 0, 1, 0, 0, 0, 0, 0, 0, 0],
              +j*(0,1,0),       [1, 0, 0, 0, 0, 1, 0, 5, -2, 0, 0, 0, 0]]
         color=5-2*i)
```

(a) Program             (b) Matrix Representation

Figure 9: Data format of our scene programs. Each color represents a different type of argument, including red: program tokens, blue: iteration arguments, green: position arguments, purple: color arguments.

