# OpenReview forum: "Learning to Describe Scenes with Programs"
_ICLR.cc/2019/Conference_

### Official Review · AnonReviewer3 · 2018-11-01

**Rating:** 6
**Confidence:** 4

**Review:**

This paper presents a system that infers programs describing 3D scenes composed of simple primitives. The system consists of three stages each of which is trained separately. First, the perceptual module extracts object masks and their attributes. The objects are then are split into several groups. Finally, each group is mapped to a corresponding DSL program using a sequence-to-sequence network similar to the ones typically employed in neural machine translation.

Pros:
+ The paper is written clearly and easy to read.
+ Visual program synthesis is very exciting and important direction both for image understanding and generation.
+ The results on synthetic datasets are good. The authors also demonstrate the applicability of the approach to real-world data (albeit significantly constrained).
+ I find it surprising that a seq2seq is good at producing an accurate program for a group of objects.
+ Visual analogy making experiments are impressive.

Cons:
- The proposed model requires rich annotation of training data since all the components of the systems are trained in a supervised fashion. It’s not clear how to use the method on the in-the-wild data without such annotation.
- Related to the previous point, even when it’s possible to synthesize data, it is non-trivial to obtain the ground-truth grouping of objects. Judging by Table 2, it seems that the system breaks in absence of the grouping information.
- The data used in the paper is quite simplistic (limited number of primitives located in a regular grid). I’m wondering if there is a natural way to extend the approach to more complex settings. My guess is that the performance will drop significantly.

Notes/questions:
* Section 2, paragraph 1: The paper by [Ganin et al., 2018] presents both a system for reproducing an image as well as for sampling from a distribution; moreover, it presents experiments on 3D data (i.e., not limited to drawing).
* Section 3.4, paragraph 2: I’m not sure I understand the last sentence. How can we know that we successfully recovered the scene at test time? Could the authors elaborate on the stopping criterion for sampling?
* Section 4.2, paragraph 2: Do I understand correctly that the main difference between the test set and the generalization set is the number of groups? (i.e., 2 vs 3). If so, it’s a fairly limited demonstration of generalization capabilities of the system.
* Section 4.2, paragraph 4: “we search top 3 proposals ...” – How do we decide which one is better? Do we somehow have an access to the ground truth program at test time?
* Could the authors explain the representation of a program more clearly? How are loops handled? How can one subtract/add programs in the analogy making experiment?

Overall, I think it is a interesting paper and can be potentially accepted on the condition that the authors address my questions and concerns.

---

> ### Author Response · Authors · 2018-11-20
> **Our Response to Reviewer 3**
>
> Thank you for your thoughtful review.
>
> 1. Data annotations
>
> Our model requires supervised training data in the pre-training stage for each of its components, but the program synthesizer needs no further training when generalizing to other data distributions since it works on object attribute space. We consider this as an advantage of our model, as it is easy to get synthetic data, but much harder to obtain annotations (in particular program annotations) on in-the-wild images. Disentangling vision recognition and program synthesis is a key to our model’s success on real images (Fig. 7): our model accurately predicts programs for the test set with only 90 labeled real images for fine-tuning.
>
> The group information is inherently included in the program and easy to obtain: when synthesizing data, we first sample several program blocks from our DSL, where each block corresponds to a set of objects that form a group. These program blocks are then combined into the overall program description of the scene. Our model explicitly learns the group information and shows advantages over baseline methods which learn directly from the overall program.
>
> 2. Scene complexity
>
> In our main experiment, we place objects on a grid and then jitter their positions and orientations. Results on these data suggest that scene programs describe these structured images well. We agree with reviewers on the importance of handling more complex data. In the revision by Nov. 26, we will add results on scenes where objects are placed at random.
>
> 3. Specific questions
>
> 1) Ganin et al: We will revise the description of this work in the updated draft.
>
> 2) The stopping criterion is whether the scene has been successfully reconstructed, i.e., when the execution results of the program are the same as the object parsing results obtained from the vision module.
>
> 3) Generalization: Yes, the main difference is the number of groups. This is only one of our experiments on generalization. The experiments on real image show our model’s ability to generalize to new data distributions.
>
> 4) Top proposals: Note that the group detector also outputs the classification result of the group category. Here “top” refers to the softmax score. We don’t have any information of the ground truth program at test time.
>
> 5) Program representations: We will give detailed definitions in the Appendix. In short, a program is represented as a matrix, where each row contains a program command, which is a program token followed by its parameters.
>
> Thanks! Please don’t hesitate to let us know for any additional comments on the paper or on the planned changes.

---

> > ### Author Response · Authors · 2018-11-27
> > **Our Response to Reviewer 3**
> >
> > Thanks again for your review. We have updated our manuscript accordingly and posted a summary of changes. Please don’t hesitate to let us know if you have additional feedback.

---

> > ### Author Response · Authors · 2018-12-04
> > **Looking forward to your feedback**
> >
> > Dear Reviewer 3,
> >
> > We would like to thank you again for your constructive review. We have revised the paper accordingly. In particular, we have included comparisons with additional, search-based baselines and increased the complexity of the scene. We have also revised the text and included a supplementary material for better clarity and more implementation details.
> >
> > As the discussion period is about to end, please don’t hesitate to let us know if there are any additional clarifications that we can offer, as we would love to convince you of the merits of the paper. We appreciate your suggestions. Thanks!

---

### Official Review · AnonReviewer2 · 2018-11-04
**good problem; weak evaluation and motivation**

**Rating:** 4
**Confidence:** 3

**Review:**

This paper investigates a descriptive representation of scenes using programs. Given an input image and an initial set of detections obtained from bottom-up detectors a sequence to sequence network is used to generate programs in a domain specific language (DSL). The authors consider a dataset where simple primitives are arranged in layouts in 3D scenes with varying material and color properties. They argue that the scene representation lead to better generalization on novel scene types and improve over baselines on image analogy tasks. The paper is well written but the evaluation and technical novelty is weak.

First, the use of scene programs is not a contribution of this paper. Going beyond the works cited in the related work section, several recent works have proposed and investigated the advantages of program synthesis for shape generation (e.g., CSGNet Sharma et al. CVPR 2018 and Scene derendering, Wu et al., CVPR 2017), visual reasoning (Modular networks, Andreas et al., 2015), among others.

At a high-level the motivation of the program level representation for the considered tasks is not highlighted. It seems that an attribute-based representation, i.e., the output of the mask R-CNN detector that describes the image as a collection of objects, material properties, and their positions and scales is a sufficient representation. The higher-order relationships can be relatively easily extracted from the detections since the images are clean and clutter free. A baseline approach where the program synthesis was performed using search and grouping should be compared with.

The considered tasks are relatively simple achieving 99.5% token-level accuracy. The evaluation beyond the synthetic datasets is fairly limited and it is unclear how well the method generalizes to novel images in clutter and occlusion.

In summary, the paper makes a number of observations that have been motivated in a number of prior works, but the contributions of this paper is not highlighted (e.g., over neural scene derendering). The main claim that higher-order relationships are being modeled is not apparent due to the simplicity of the scenes being considered. For example, the program blocks being considered are somewhat arbitrary and a comparison with a clustering based grouping approach should have been evaluated. The experimental evaluation is weak in several aspects. The generalization to real images is anecdotal with only two examples shown in the Figure 7.

---

> ### Author Response · Authors · 2018-11-20
> **Our Response to Reviewer 2**
>
> Thank you for your thoughtful review.
>
> 1. Scene program as a contribution
>
> Thanks for suggesting the related work, which we’ll cite and discuss. Our paper is new and different from all these papers. Most importantly, our scene programs focus on modeling the high-level structural relationships among multiple objects, building upon and extending CSGNet and De-rendering, which only explored sequential, primitive-level programs. Only with our scene programs and the loop structure, may we efficiently describe patterns that involve higher-order, program-like regularity among multiple objects (e.g., repetition) and edit images with minimal interactions (Fig. 6,7).
>
> 2. High-level motivations
>
> As mentioned above, many tasks become possible only with our program representations, not an attribute-based representation. For example, for the image in Fig. 6(a), it is natural for one to imagine how could we add another row of cylinders on the right side, or change cylinders to cubes. Scene programs allow us to perform such tasks in an efficient and intuitive way. However, if we represent the scene by its objects and attributes, we would have to individually edit each object, losing the benefit from their high-level correlations.
>
> We agree that it’s important to discuss the alternative approach, where the program synthesis is performed using search and grouping. There are two possible approaches for a structured search over the space of programs, both of which would be too slow for our task:
> - Constraint solving: we would have to use an SMT solver. Ellis et al [1] used SMT solvers to infer 2D graphics programs, and takes on the order of minutes per program. As 3D scenes (with attributes) have a much larger search space, such an approach would not be able to find a solution in reasonable time.
> - Stochastic search: Here the problem would be at least as tough as doing inverse graphics, so we can safely assume that this would work no better than MCMC for inverse graphics. In Picture (Kulkarni et al. [2]), their approach takes minutes for a 2D image with simple contours.
>
> We have contacted the authors of these two papers, who confirmed our estimates of the efficiency of their methods. In comparison, on average our neural program synthesis model takes less than 0.4 second to generate scene programs for an image.
>
> [1] Ellis, Kevin, Armando Solar-Lezama, and Josh Tenenbaum. "Unsupervised learning by program synthesis." NIPS 2015.
> [2] Kulkarni, Tejas D., et al. "Picture: A probabilistic programming language for scene perception." CVPR 2015.
>
> 3. Experimental evaluation
>
> In our main experiment, we place objects on a grid and then jitter their positions and orientations. Results on these data suggest that scene programs describe these structured images well. We agree with reviewers on the importance of handling more complex data. In the revision by Nov. 26, we will add results on scenes where objects are placed at random.
>
> Following your suggestion, we have also experimented with a simple heuristic grouping algorithm: we start with a random object which forms a group. As long as there are objects of distance less than a threshold from this group, we add them into the group. It performs well in general, but cannot resolve difficult instances, such as when a group is surrounded by other objects. This justifies the need to use a deep model to learn from the data distribution. We’ll include the results in the revision.
>
> To summarize, our main contribution is to propose scene programs as a novel representation of scenes, modeling high-level relations beyond individual objects and pairwise relations. The scene program representation demonstrates strong advantages in tasks such as image editing and analogy making, compared with attribute-based representations. We also propose effective methods to generate accurate program descriptions from input images, which can be applied to real images without additional program supervision. While tackling in-the-wild scenes is an important future direction, we consider our efforts an important step towards high-level scene understanding via program synthesis.
>
> Please don’t hesitate to let us know for any additional comments on the paper or on the planned changes.

---

> > ### Author Response · Authors · 2018-11-27
> > **Our Response to Reviewer 2**
> >
> > Thanks again for your review. We have updated our manuscript accordingly and posted a summary of changes. Please don’t hesitate to let us know if you have additional feedback.

---

> > ### Author Response · Authors · 2018-12-04
> > **Looking forward to your feedback**
> >
> > Dear Reviewer 2,
> >
> > We would like to thank you again for your constructive review, which has helped us improved the quality of the paper. We have cited and discussed the related work as suggested. We have also included comparisons with additional, search-based baselines and increased the complexity of the scene.
> >
> > As the discussion period is about to end, please don’t hesitate to let us know if there are any additional clarifications that we can offer, as we would love to convince you of the merits of the paper. We appreciate your suggestions. Thanks!

---

### Official Review · AnonReviewer1 · 2018-11-05
**A novel scene representation proposed, but needs more clarification on its advantages and flexiblity.**

**Rating:** 6
**Confidence:** 3

**Review:**

[Overview]

In this paper, the authors proposed a new format of representation called scene programs, to describe the visual scenes. To extract the scene programs from scenes, the authors exploited the off-the-shelf object detection and segmentations model, mask r-cnn to extract all objects and the corresponding attributes from the images, and then detect groups for those objects, which are then used to generate the programs which matches the input scenes. The experiments are performed on a synthetics datasets which consists of multiple shapes with different attributes. The experiments shows that the proposed model can infer more accurate programs from the scenes, and those generated programs can be used to recover the input scenes more accurately. Besides, the authors also showed that the generated scene programs can be used for image editing and making visual analogy.

[Strengthes]

1. The authors proposed a new representation, called scene programs, to describe the visual scenes with some textual program. This is a new scene representation, which could be potentially used in various scenarios, such as the image synthesis in graphics.

2. The authors proposed a hierarchical method to model the structures in scenes. Specifically, the objects in a scene are first extracted and then grouped into multiple clusters, which will be used to guide the scene program synthesis.

3. The experimental results demonstrate the effectiveness of the proposed method both qualitatively and quantitatively. The authors also showed the the programs generated  can be sued for image editing and cross-modality matching.

[Weaknesses]

1. It is a bit unfair to compare the proposed method with the two baseline methods listed in Table 2. The authors used a pre-trained mask-rcnn to detect all objects and predict the attributes for all objects. However, the counterpart methods have no access to this supervision. Even in this case, CNN-LSTM seems achieve comparable performance on the first three metrics.

2. The advantage of scene program compared with scene graph (Johnson et al) are not clear to me. Scene graph is also a symbolic representation for images. Also, for all the tasks mentioned in this paper, such as image editing and visual analogy, scene graph can probably also complete well. The authors should comment about the specific advantages of scene program in comparison with scene graph.

3. All the images shown in the paper seems arranged uniformly, which I think contains some bias to the proposed grouping strategy. I would like to see more diverse configurations of the foreground objects. It would be good to see if the proposed model can describe more complicated scenes.

[Summary]

This paper proposed a novel scene representations, called scene program. To extract the scene program, the authors proposed a hieratchical inference method. The resulting scene programs based on the proposed model outperforms several baseline models quantitatively. The authors also showed the proposed scene program is suitable for image editing and visual analogy making. However, as pointed above, there are some unclear points to me, especially the advantages of scene program compared with scene graph, and the representation power of scene program for complicated scenes.

---

> ### Author Response · Authors · 2018-11-20
> **Our Response to Reviewer 1**
>
> Thank you for your thoughtful review.
>
> 1. Comparing with baseline methods
>
> We agree on the importance of a fair comparison. We’d like to clarify that
> 1) The derender-LSTM baseline requires the same supervision as ours.
> 2) We acknowledge the fact that our method requires more supervision than the CNN-LSTM baseline method, and will revise the paper to clearly state that. Note the situation is different when we consider generalization to data where program supervision is hard to obtain (e.g. real images). Because our program synthesizer works on abstract object attribute space, it does not require any fine-tuning to work on real images, where the end-to-end CNN-LSTM approach would require (image, program) pairs. Disentangling vision recognition and program synthesis is a key to our model’s success on real images (Fig. 7).
>
> Our method achieves higher test accuracy and better generalization performance than both baseline, and more importantly, it generalizes better to real images. As synthetic data can be easily obtained, we believe that the comparison in Table 2 is fairly established. These results and the experiments on real images demonstrate the significant advantage of our method.
>
> 2. Scene graphs
>
> We thank the reviewer for bringing up scene graphs, which is another important high-level representation of scenes. Here we would like to emphasize that the roles of scene graphs and the proposed scene programs are complementary: scene graphs focus on the pairwise relationships, e.g. an object can be on, in, or under another object; in contrast, scene programs focus on the higher-order, program-like regularity among multiple objects (e.g., repetition).
>
> Compared with scene graphs, scene programs (i) explicitly capture these regularities, modeling correlations among multiple objects; and (ii) are more efficient in terms of description length. For tasks such as editing structured images, scene programs are a more suitable representation, because editing can be performed on the program space for higher efficiency, as displayed in Fig. 6. With scene graphs, we would have to edit each object one by one.
>
> 3. Scene complexity
>
> In our main experiment, we place objects on a grid and then jitter their positions and orientations. Results on these data suggest that scene programs describe these structured images well. We agree with reviewers on the importance of handling more complex data. In the revision by Nov. 26, we will add results on scenes where objects are placed at random.
>
> Please don’t hesitate to let us know for any additional comments on the paper or on the planned changes.

---

> > ### Author Response · Authors · 2018-11-27
> > **Our Response to Reviewer 1**
> >
> > Thanks again for your review. We have updated our manuscript accordingly and posted a summary of changes. Please don’t hesitate to let us know if you have additional feedback.

---

> > ### Author Response · Authors · 2018-12-04
> > **Looking forward to your feedback**
> >
> > Dear Reviewer 1,
> >
> > Thanks again for your constructive review, which has helped us improved the quality and clarity of the paper. In addition to our response above, in the revision, we have included comparisons with additional baselines and increased the complexity of the scene.
> >
> > As the discussion period is about to end, please don’t hesitate to let us know if there are any additional clarifications that we can offer, as we would love to convince you of the merits of the paper. We appreciate your suggestions. Thanks!

---

### Public Comment · (anonymous) · 2018-11-22
**Interesting work that aims to describe real-world scenes with programs**

This visual program synthesis work is really interesting. However, some of the related work is missing, including:
*describe scenes with programs*
pix2code: Generating Code from a Graphical User Interface Screenshot
*describe the Structure of 3D shapes*
Learning Shape Abstractions by Assembling Volumetric Primitives
*program synthesis*
Neural Program Synthesis from Diverse Demonstration Videos,
Leveraging Grammar and Reinforcement Learning for Neural Program Synthesis

---

> ### Author Response · Authors · 2018-11-27
> **Thank you for your interest in our work**
>
> Thanks for your comments! We have cited and discussed these related works in our revised manuscript.

---

### Author Response · Authors · 2018-11-27
**Our Revised Manuscript**

We thank all reviewers for their helpful comments. We have performed additional experiments and revised the manuscript according to the reviewers’ suggestions, The specific changes include:

1. We have cited and discussed additional related work in Sec. 2.

2. We have added a search-based heuristic grouping method as an ablated version of our full model, with descriptions in Sec.
4.2 (paragraph 1) and quantitative results in Table 2.

3. We have added results on a new, more challenging dataset with random object arrangement. The updated quantitative results are in Table 2. We have also added relevant discussions in Sec. 4.1 (paragraph 2) and 4.2 (setup & results).

4. We have included more qualitative results on both synthetic and real images in Fig. 3c and Fig. 7.

5. We have included an appendix, summarizing implementation details on scene configuration (synthetic data generation), the heuristic grouping baseline, and data formats.

---

### Comment · Area_Chair1 · 2018-11-30
**Reviewers take notice**

Thank you all for your substantial reviews. As you can see, the authors have responded with detailed comments and rebuttals of their own, and revised the paper. It is imperative that you examine these rebuttals, the revisions made to the paper, and each others reviews. Please engage in further discussion where needed, but begin to either reconsider your assessment, or formulate an explanation as to why you are standing by your scores.

---

### Public Comment · ~Douglas_Blank1 · 2019-05-08
**Related work**

For an early example of a neural network used in visual analogy making, please see:

Blank, D. 1997. Learning to See Analogies: A Connectionist Exploration. PhD. thesis. Indiana University. https://repository.brynmawr.edu/cgi/viewcontent.cgi?article=1077&context=compsci_pubs

---

### Meta-Review · Area_Chair1 · 2018-12-14
**Acceptable**

**Confidence:** 3
**Recommendation:** Accept (Poster)

**Metareview:**

This paper presents a dataset and method for training a model to infer, from a visual scene, the program that would generate/describe it. In doing so, it produces abstract disentangled representations of the scene which could be used by agents, models, and other ML methods to reason about the scene.

This is yet another paper where the reviewers disappointingly did not interact. The first round of reviews were mediocre-to-acceptable. The authors, I think, did a good job of responding to the concerns raised by the reviewers and edited their paper accordingly. Unfortunately, not one of the reviewers took the time to consider author responses.

In light of my reading of the responses and the revisions in the paper, I am leaning towards treating this as a paper where the review process has failed the authors, and recommending acceptance. The paper presents a novel method and dataset, and the experiments are reasonably convincing. The paper has flaws and the authors are advised to carefully take into account the concerns flagged by reviewers—many of which they have responded to—in producing their final manuscript.